# Corporate financial distress prediction with multiperiod annual report data: A fusion deep neural network model

Chongren Wang[1,2], Pimei Gong [1,2]*, Jiawang Li[1], Zhiyi Wang[1,2]

**1** School of Management Science and Engineering, Shandong University of Finance and Economics, Jinan, China, **2** Digital Economy Research Institute, Shandong University of Finance and Economics, Jinan, China

\* m1926017049@gmail.com

## Abstract

The occurrence of financial distress in enterprises not only leads to operational difficulties but also may trigger chain reactions such as bankruptcy, debt arrears, layoffs, etc., which in turn have a negative effect on investors, creditors, and the entire economic system. Therefore, accurately and timely predicting the financial distress of enterprises is highly important. To address this, a fusion deep neural network based on multiple annual report text data and financial data (MTF-FDNN) model is proposed for financial distress prediction. This model can simultaneously extract long text features of multiple annual reports and financial indicator features of multiple periods of enterprises. Specifically, the model first constructs a multiperiod financial feature extraction model on the basis of a fully connected neural network. Next, it uses a fine-tuned longformer pretrained model to convert long texts into vector representations. Subsequently, Bi-LSTM and TextCNN are employed to extract semantic features from long texts both globally and locally. Finally, the fused financial features and semantic features of long texts are used to identify the financial distress of listed companies. Additionally, on the basis of the experimental results from the test set, the proposed model demonstrates significant improvements over traditional multiperiod financial indicator-based prediction models, with increases of 4.98% in AUC, 6.54% in accuracy, 10.58% in recall, and 6.48% in the F1 score. It is evident that introducing multiperiod textual features significantly enhances model predictive performance. This model effectively predicts corporate financial distress, thereby assisting business managers, external investors, and other stakeholders in mitigating risk.

## Introduction

With the rapid development of the global economy and intensified market competition, the financial risks faced by enterprises have become increasingly complex and

**Data availability statement:** The datasets generated and analyzed during this study are available in the CSMAR repository (https://data.csmar.com/). Researchers can obtain financial indicator data from the "Financial Indicator Analysis" section under the "Corporate Research" series in the CSMAR database, and access annual report texts and label data from the "Operational Distress" section, after registration.

**Funding:** This study was supported by the Shandong Province Science and Technology-based Small and Medium-sized Enterprises Innovation Capacity Enhancement Project (2023TSGC0208) and the Jinan City School Integration Development Strategy Project (JNSX2023052). The funders had no role in study design, data collection and analysis, decision to publish, or preparation of the manuscript.

**Competing interests:** The authors have declared that no competing interests exist.

diverse, making them more prone to financial distress. Corporate financial distress is a typical financial risk that indicates a higher probability of financial crisis and bankruptcy. The occurrence of financial distress not only leads to operational difficulties for the enterprise itself but also may trigger a series of chain reactions, such as bankruptcy, debt default, and layoffs. These consequences not only cause direct losses to investors and creditors but also may have widespread adverse impacts on the entire economic system. Therefore, accurately and promptly predicting corporate financial distress has become a focal point of interest for both academics and practitioners. Financial distress prediction (FDP) can assist corporate management in identifying potential issues early and taking preventive measures while also providing risk warnings to investors and creditors, thereby optimizing resource allocation. Moreover, governments and regulatory agencies can leverage these predictions to formulate effective policies to reduce systemic risk and maintain the stability of financial markets. Hence, research on financial distress prediction holds significant theoretical and practical value. The key challenge in the current field of financial risk management research is how to fully and effectively utilize both financial and nonfinancial information disclosed by enterprises to identify and predict financially distressed firms accurately.

The prediction of financial distress can be traced back to univariate discriminant analysis. Beaver W.H. et al. [1] utilized financial ratios to predict corporate financial distress and verified the value of different types of data in financial statements for financial distress prediction. However, a single indicator fails to comprehensively reflect a company's overall financial condition, and the predictive effectiveness of different financial ratios varies significantly. Therefore, research has gradually shifted toward multivariate analysis [2–4]. Although multivariate analysis has improved prediction accuracy to some extent, traditional methods mainly rely on single-period financial data, which only reflect a company's operational status at a specific point in time and fail to capture the long-term evolution of its financial condition. Consequently, research has gradually expanded toward the analysis of multi-period financial indicators to more comprehensively reveal the dynamic changes in corporate financial distress [5]. In recent years, the rapid development of text mining techniques has provided new tools for traditional accounting and financial research, driving innovations in financial distress prediction methods. Scholars have begun to introduce textual data disclosed in corporate annual reports to extract valuable insights into corporate operations [1,6], while also exploring the integration of financial data and textual data to enhance the predictive ability and accuracy of models [7]. In terms of model construction, traditional statistical methods, such as the univariate discriminant model proposed by Beaver [1], the multiple discriminant analysis (MDA) model proposed by Altman [8], and the logistic regression (LR) model proposed by Ohlson [2], have been widely applied. Among them, the classic Altman Z-Score model has achieved significant success in corporate bankruptcy risk prediction. With the advancement of research, machine learning models have gradually demonstrated stronger predictive capabilities by providing more flexible nonlinear mappings. In particular, artificial neural networks (ANN) [9] and support vector machines (SVM)

[5,10,11] have been widely used in financial distress prediction. In recent years, the rapid development of deep learning techniques has further improved the accuracy of financial distress prediction [12,13]. By constructing complex deep neural network models, researchers have been able to more effectively extract deep features from multi-period financial data and textual data, thereby enhancing prediction precision and robustness. However, existing studies have the following limitations: First, textual feature extraction methods are relatively simple, as Convolutional Neural Network (CNN) or Long Short-Term Memory (LSTM) are commonly used, failing to effectively combine both global and local textual features. Second, most studies only utilize single-period annual report text, without considering the long-term dynamic evolution of textual information. Third, multimodal data fusion remains insufficient, as most studies focus either on financial indicators or textual data, lacking a joint modeling approach for multi-period financial indicators and multi-period annual report text.

To address the above issues, this study proposes a financial distress prediction model that integrates multi-period financial indicators and multi-period annual report text to improve prediction accuracy and stability. First, to extract features from multi-period annual report text, a novel neural network architecture is designed to fully capture textual semantic information. Longformer is adopted due to its sliding window and global attention mechanism, which has low computational complexity and can effectively capture long-range dependencies in long financial reports. Bidirectional Long Short-Term Memory (Bi-LSTM) is used to extract global textual features, as its bidirectional information flow captures contextual dependencies, making it suitable for financial report texts with strong sequential relationships. Meanwhile, Text Convolutional Neural Network (TextCNN) extracts local features through different convolutional kernels, capturing phrase-level information. The combination of Bi-LSTM and TextCNN allows the model to incorporate both global and local textual features.

Therefore, this study employs Longformer to extract deep semantic information from the text, Bi-LSTM to extract global features, and TextCNN to extract local features. Subsequently, a dynamic financial distress prediction model based on multi-period annual report data is constructed. Finally, multi-period textual semantic features and multi-period financial features are jointly modeled and fused using neural network-based fusion techniques to improve the accuracy of financial distress prediction. The main contributions of this study are as follows:

(1) Development of a feature fusion model to extract semantic features from long texts.

This study applies multiperiod financial report texts to the field of financial distress prediction and designs a novel neural network architecture tailored to the characteristics of multiperiod financial report texts. This approach enhances the prediction performance of financial distress models that incorporate long texts from multiperiod financial reports. First, the pretrained Longformer model is utilized to compute word vectors for the long texts in financial reports. Subsequently, the global features of the text were extracted via Bi-LSTM, and the local features were extracted via TextCNN. A feature-level fusion method is then used to construct multiperiod semantic features of long texts, offering a new perspective for extracting semantics from multiperiod financial report texts.

(2) A dynamic financial distress prediction model is constructed on the basis of multiperiod annual report data.

A dynamic financial distress prediction model is constructed on the basis of multimodal data from multiperiod annual report texts and financial indicators. First, multiple fully connected neural networks are used to process multiperiod financial indicator data. Next, a fusion model is designed to handle multiperiod annual report texts by extracting long texts from the Management Discussion and Analysis (MD&A) sections of multiple years. The feature fusion model is then used to compute the global and local semantic features of these long texts, and the results are integrated as multiperiod text semantic features. Finally, a neural network combines the extracted multiperiod textual semantics and multiperiod financial features for predictive computation.

This study is structured as follows. First, the current research on financial distress prediction based on financial indicator data, annual report text data, and the combination of the two is reviewed. Then, the proposed financial distress prediction model is described in detail, including the problem definition, overall framework, and model architecture. Next, the experimental setup, dataset processing, evaluation metrics, comparison with existing methods, ablation studies, and

a comprehensive analysis and discussion of the results are introduced. Finally, the limitations of the current research are discussed and directions for future research are outlined.

## Related work

The main objective of financial distress prediction is to identify the financial health of a company in advance and help managers, investors, and other stakeholders take measures to address potential issues. Common financial distress prediction methods can be categorized into three types: methods based on financial indicator data, methods based on annual report text data, and methods that combine annual report text and financial indicator data.

### Financial distress prediction based on financial indicators

In recent years, significant progress has been made in the prediction of financial distress based on financial indicator data, with numerous scholars employing various methods to improve prediction accuracy and stability. Traditional statistical methods still play an important role in this field. For example, Juliana Adeola Adisa et al. [14] used linear discriminant analysis (LDA) to optimize the classic Z-Score model, and combined it with logistic regression to construct a corporate TDR probability index, improving the classification of distressed and non-distressed companies. Additionally, Malakauskas Aidas et al. [15] used machine learning methods, including logistic regression, artificial neural networks (ANN), and random forests, to construct a binary classifier based on data from 12,000 small and medium-sized enterprises. They also incorporated factors such as time, credit history, and company age into classic financial ratios, which improved prediction accuracy. In recent years, machine learning and deep learning methods have become increasingly prevalent in financial distress prediction. Zhang Rui et al. [16] used 14 financial indicators from two years prior ($t-2$) to predict whether a company would receive an ST tag in the current year ($t$), proposing a GA-MLP model that utilizes genetic algorithms (GA) to optimize multilayer perceptrons (MLP) for enhanced prediction performance. Similarly, Lei Ruan et al. [17] used genetic algorithms to optimize backpropagation (BP) neural networks, overcoming the slow convergence speed and tendency to fall into local optima in BP networks, thereby improving prediction accuracy and stability. Kennedy et al. [18] constructed a predictive model using artificial neural networks (ANN) and tested it on a sample of 22 distressed companies and 33 non-distressed companies, achieving a prediction accuracy of 85.7%. In addition to individual machine learning models, ensemble learning and hybrid models have also shown excellent performance in financial distress prediction. Michal et al. [19] selected 27 financial variables and used RobustBoost, CART, and optimized k-NN for training and validation. The results indicated that the hybrid model outperformed individual models. At the same time, Yang Dawen et al. [20] extended traditional financial ratios by incorporating lagging financial ratios from 3 to 5 periods and macroeconomic factors, constructing a predictive model using lasso penalty logistic regression and SVM lag framework. Their results showed that compared to a single time-window model, a model that integrates indicators from multiple time windows conveys more information, thereby improving the stability and reliability of financial distress predictions. Overall, financial distress prediction research has gradually evolved from traditional statistical methods to machine learning, deep learning, and ensemble learning approaches. In the future, more complex model structures can be explored to enhance prediction capability.

In summary, current research mainly focuses on introducing more advanced algorithms and more complex features to improve prediction performance. However, these methods still face certain limitations. First, while traditional financial ratios have some predictive power, relying solely on financial indicators may overlook other important factors of a company, such as governance structure, industry background, and more. Second, although ensemble learning and deep learning methods have been widely applied, the complexity of these models and their high computational requirements remain bottlenecks in practical applications. Future research can further explore the integration of more non-financial data or develop more efficient feature extraction methods to improve prediction performance.

## Financial distress prediction based on annual report text data

Financial distress prediction based on textual data has become an important research direction in the financial field, with financial text mining widely regarded as a powerful tool to address this issue. Shixuan Li et al. [15]proposed a deep learning-based framework that, by constructing a sentiment lexicon specific to the Chinese financial sector and combining word vector models with deep learning classifiers, demonstrated that sentiment features derived from a financial domain lexicon exhibited clear advantages over other sentiment lexicons in financial distress prediction. Huang Bo et al. [21] extracted sentiment information from the Management Discussion and Analysis (MD&A) and audit reports in Chinese-listed companies' annual reports and used deep learning algorithms for sentiment analysis. The results indicated that incorporating sentiment scores into the model significantly improved prediction performance. Additionally, Cuiqing Jiang et al. [22] employed word embedding techniques to extract semantic features, with empirical research showing that compared to periodic reports, more valuable information in current reports helped predict the financial distress of unlisted public companies. The prediction performance was significantly improved using the extracted semantic features, surpassing topic features and sentiment features. Shuping Zhao et al. [23] combined sentiment features from online stock forum comments, MD&A, and financial statement footnotes, utilizing the CatBoost model for financial distress prediction. Their results showed that including textual data effectively identified more distressed companies.

In the application of financial sentiment analysis, Hajek, Petr et al. [24] utilized a BERT-based contextual embedding model, focusing on language analysis of risk-related sections in corporate annual reports, extracting detailed financial sentiment and thematic coherence. Experimental results showed that this model significantly outperformed most existing financial distress prediction models in prediction accuracy. Zi Nie et al. [25] investigated the impact of annual report disclosure delays on financial distress prediction accuracy, finding that delayed information disclosure significantly improved prediction accuracy. Further, Song Yang et al. [26] examined the impact of ESG (Environmental, Social, and Governance) performance on the financial distress risk of companies in the energy sector. By applying a triangular approach of sentiment, topic, and word frequency analysis to extract ESG report text features, and integrating them with other variables such as company carbon performance, they incorporated these into the CatBoost model. Their empirical results demonstrated that ESG-related textual variables effectively enhanced the accuracy of financial distress predictions.

In summary, financial text mining, particularly sentiment analysis and semantic feature extraction, has become an important method for improving the accuracy of financial distress prediction. Future research can further explore multi-modal data fusion, integrating financial reports, news articles, social media, and other multi-source textual information to further enhance model prediction capabilities. Significant progress has been made in financial distress prediction based on textual data, particularly in sentiment analysis and semantic feature extraction, with the application of deep learning frameworks and diverse text features (e.g., MD&A sections, ESG reports) greatly enhancing prediction accuracy. However, existing methods still face some challenges. For example, while the potential for multimodal data fusion is enormous, effectively integrating data from different sources, such as financial statements, news reports, and social media, remains a critical issue that needs to be addressed. Future research could combine various sentiment analysis techniques, such as contextual sentiment analysis and deep semantic understanding, while enhancing the fusion of cross-domain textual data to improve the accuracy and stability of the models.

## Financial distress prediction combining annual report text and financial indicator data

In recent years, the integration of financial indicators and textual data for financial distress prediction has gradually become an important research direction, achieving significant results. Tang Xiaobo et al. [7] proposed a comprehensive model that combines financial, managerial, and textual factors. They employed a wrapper-based feature selection method to extract valuable features and built multiple single classifiers, ensemble classifiers, and deep learning models for financial distress prediction. The experimental results demonstrated that managerial and textual factors effectively supplement

traditional financial indicators, particularly the inclusion of textual factors, which significantly enhanced model performance. This finding aligns with the research of Shixuan Li et al. [27], who integrated financial data with multiple types of textual prediction factors. They used frequency counting, TF-IDF, TextRank, and word embedding methods to extract features based on frequency counts, keywords, sentiment, and readability metrics. Their research showed that sentiment dictionaries specific to the financial domain, readability analysis methods based on word embeddings, and fundamental textual features from the Management Discussion and Analysis (MD&A) section are crucial for improving the accuracy of financial distress prediction. Ying Chen et al. [28] applied L1/2 regularization to constrain the weights of hidden layers, selecting both financial and non-financial variables, which significantly improved prediction accuracy. Non-financial variables, particularly those related to investor protection and corporate governance, were found to play a more critical role in financial distress prediction than traditional financial variables. Furthermore, M. Ragab Yasmin et al. [29] explored the impact of governance-related non-financial variables on financial distress prediction for SMEs in Egypt's listed companies. Their results indicated that models combining financial and governance-related non-financial variables achieved a 1.9% higher prediction accuracy compared to models using only financial variables. Further research has also explored the potential of extracting information from non-traditional data sources. For instance, Cui qing Jiang et al. [30] investigated whether interactive and thematic features extracted from Q&A texts could significantly enhance the performance of financial distress prediction (FDP) models. Their results demonstrated that models incorporating interaction and thematic features exhibited a substantial improvement in prediction accuracy.

The integration of financial indicators and textual data has proven to be a powerful approach, particularly with the introduction of non-financial factors and sentiment analysis. However, such models still face challenges, primarily in the complexity of data processing and the high computational demands of model training. Future research could focus on how to efficiently integrate multi-source information, reduce redundant data, and improve model interpretability. Additionally, effectively combining non-financial variables with financial indicators, especially in terms of adaptability across different markets and industry contexts, remains a topic worthy of further exploration.

## MTF-FDNN

**Problem definition.** This study constructed a fusion deep neural network model based on multiple annual report text data and financial data (MTF-FDNN) for financial distress prediction, which is capable of leveraging the dynamic features of multiperiod data from publicly listed companies and simultaneously extracting long-text features from multiperiod annual reports and financial features over multiple periods. Related research has shown that the predictive effectiveness of financial distress models generally spans up to three years, with accuracy significantly declining beyond this period [31]. Publicly listed companies release their annual reports for year T-1 during year T. Following the release of these reports, the stock exchange and the regulatory agencies for listed companies often take special measures (STs) for companies in financial distress. If the data from the year immediately preceding the ST designation are used to predict financial distress in the same year, the model's predictive ability may be overstated [32]. Therefore, this study identifies the year a company is labeled as ST as the financial distress year T and uses financial indicators and annual report text data from two years T-2, three years T-3, and four years T-4 prior to the distress year to comprehensively predict financial distress.

## Overall process framework

The overall framework of the MTF-FDNN model is illustrated in Fig 1, consists of four stages.

## Financial feature extraction phase

In this phase, a fully connected neural network is employed to process the multiperiod financial features. After cleaning, filling in missing values, and standardizing the financial indicator data from the T-2, T-3, and T-4 fiscal years, the data

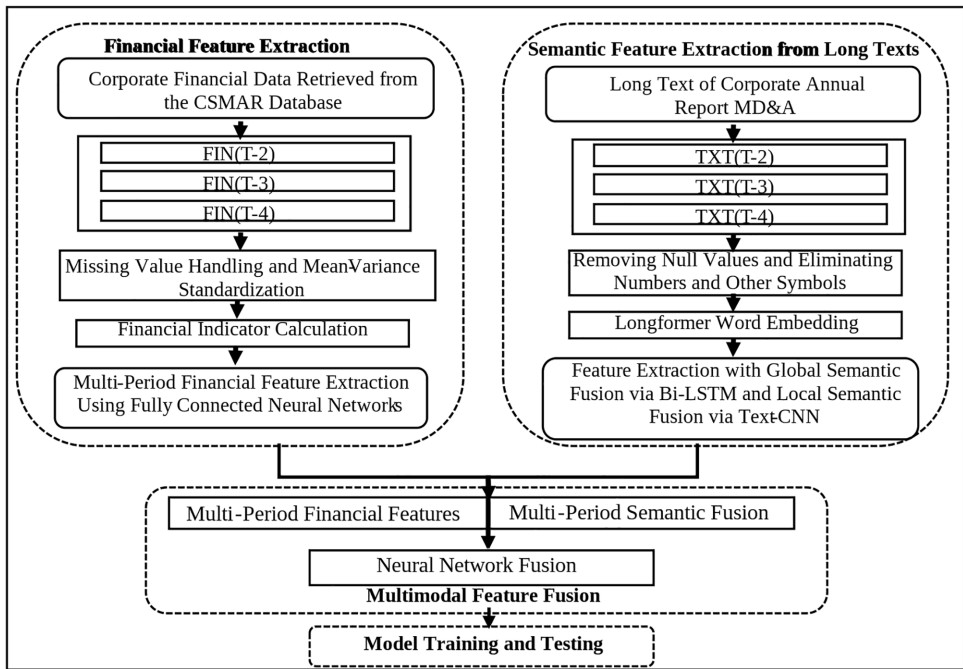

**Fig 1. Overall process framework.**

are input into separate fully connected networks to extract features. The extracted features from the three years are then fused and passed through another fully connected layer to obtain the multiperiod financial features for subsequent fusion computations.

## Long text semantic extraction phase

The MD&A long texts from the annual reports of the T-2, T-3, and T-4 fiscal years are retrieved. After preprocessing, the texts are truncated to a fixed length and sequentially concatenated into a threefold fixed-length multiperiod text dataset. Initially, a fine-tuned longformer pretrained model is used as the text embedding layer to compute semantic vectors. The Bi-LSTM model is subsequently employed to extract global semantic feature vectors, whereas the TextCNN is used to extract local semantic feature vectors. These two types of semantic feature vectors are 21 and then concatenated and passed through a neural network for feature fusion, ultimately producing multiperiod text semantic feature vectors.

## Multimodal feature fusion phase

After the multiperiod financial features and multiperiod text semantic features are extracted, a neural network model is used for feature fusion. This step not only combines the financial features and text semantic features but also integrates their respective multiperiod dynamic features to classify the financial distress of listed companies.

## Model training and testing phase

Using the available data, the extracted financial indicators and MD&A long texts from the annual reports serve as the inputs to the model, with the financial distress status of the company as the label. The model is trained with these data, and then the trained model is used to predict the financial distress of new data using the same inputs, allowing for testing of the model's prediction performance.

## Model structure

The model structure proposed in this paper is shown in Fig 2, which consists of three parts: financial feature extraction, text feature extraction, and feature fusion. It can be decomposed into five modules:

### Fully connected neural network based on financial indicator data

Using the financial status of each enterprise in the T-th fiscal year as the model label, the financial data from the T-2, T-3, and T-4 fiscal years are selected as the input to the model. Through a fully connected layer, the input financial data are mapped into a new feature space, as described in Eq. (1):

$$z_t^{(1)} = W_1^t x_t + b_1^t \tag{1}$$

where $x_t$ represents the input financial data for year $T-t$ and where $W_1^t$ and $b_1^t$ are the weight matrix and bias vector for year $T-t$, respectively.

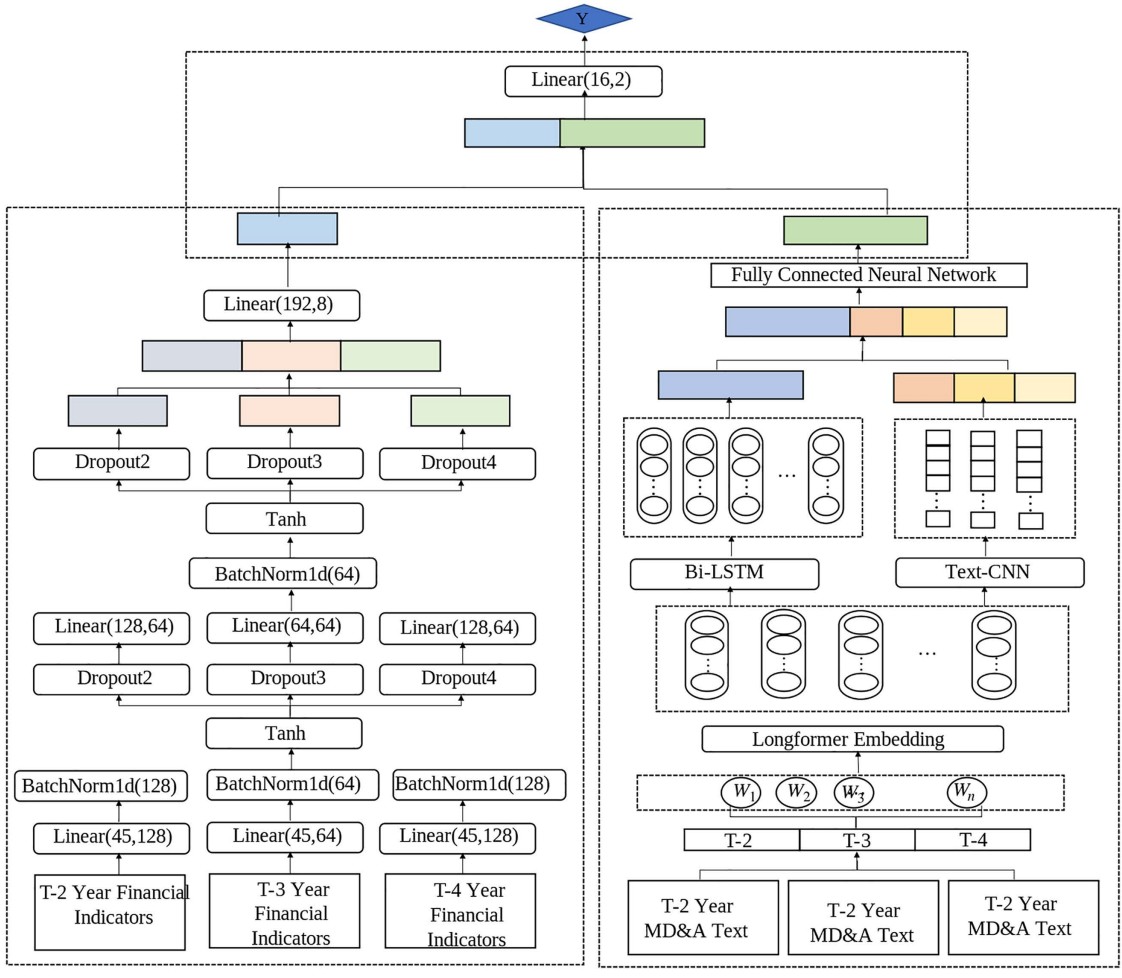

**Fig 2. Model structure.**

To improve the model's generalization ability, avoid overfitting, and accelerate the training process, the output of the linear transformation is batch normalized. Then, the tanh activation function is applied to map the data into a new range from −1--1, enabling the model to handle nonlinear relationships. Finally, dropout regularization is applied, randomly dropping a portion of the neurons with a certain probability to reduce coadaptation among neurons and enhance the model's generalization ability. The final feature extraction result from the first layer is expressed as shown in Eq. (2).

$$o_t^{(1)} = Dropout\left(Tanh\left(BatchNorm\left(z_t^{(1)}\right)\right)\right) \tag{2}$$

Similar to the first layer, the output for each year $o_t^{(1)}$ will undergo another linear transformation, batch normalization, an activation function, and dropout to obtain deeper-level features, as expressed in Eq. (3):

$$o_t^{(2)} = Dropout\left(Tanh\left(BatchNorm(W_2^t o_t + b_2^t)\right)\right) \tag{3}$$

where $W_2^t$ and $b_2^t$ are the weight matrix and bias vector, respectively, for the second layer of feature extraction for year $T-t$.

The features processed by two layers for each year are concatenated along a specific dimension, resulting in a comprehensive feature vector containing information from all years, as described in Eq. (4):

$$outcombined = concat\left(o_2^{(2)}, o_3^{(2)}, o_4^{(2)}\right) \tag{4}$$

Finally, the concatenated features are passed through a fully connected layer for classification, outputting the final multiperiod financial features, as illustrated in Eq. (5):

$$out_{final} = W_{final}outcombined + b_{final} \tag{5}$$

## Long text embedding based on longformers

The BERT model has a limitation on the input text sequence length, which cannot exceed 512 tokens. It is commonly used for research on short texts such as dialogs and comments. Beltagy et al. [22] proposed the longformer model in 2020, which can handle long texts. The number of processed MD&A texts from annual reports typically ranges from 1,000–2,000 words. To address the input length limitation of word embedding models, the fusion model developed in this study uses the encoder from the Longformer model to perform text embedding.

The encoder from the Longformer model is an enhancement of the encoder in the standard transformer, retaining the original multihead self-attention mechanism while incorporating local window attention and global attention to handle ultralong texts. In the semantic extraction of long MD&A texts, the longformer uses local window attention to capture local contextual information in the text, applies global attention to capture long-range semantic relationships across the document, and combines multihead self-attention for multiangle feature extraction, ultimately generating a rich contextual vector.

This module finally outputs an MD&A text vector sequence X, which represents the semantics of the MD&A text and provides high-quality input for subsequent tasks.

## Global semantic extraction based on Bi-LSTM

The MD&A text vectors output by the Longformer are fed as inputs to the Bi-LSTM to extract feature information from the sequence in both forward and backward directions. The Bi-LSTM consists of two LSTM layers, processing the time series

in the forward and backward directions. The forward LSTM reads the input MD&A text sequence X and sequentially computes the hidden states at each time step, as shown in Eq. (6):

$$\vec{h}_t = LSTM\left(x_t, \vec{h}_{t-1}, \vec{c}_{t-1}\right)$$

(6)

where $\vec{h}_{t-1}$ represents the previous hidden state, $\vec{c}_{t-1}$ represents the previous memory state, and $x_t$ is the input token at time step $t$.

The backward LSTM reads the input sequence in reverse order, from $x_t$ to $x_1$, and computes the backward hidden states $\overleftarrow{h}_t$, as shown in Eq. (7):

$$\overleftarrow{h}_t = LSTM\left(x_t, \overleftarrow{h}_{t-1}, \overleftarrow{c}_{t-1}\right)$$

(7)

At each time step $t$, the Bi-LSTM concatenates the forward and backward hidden states to form the feature representation at that time step, as shown in Eq. (8):

$$h_t = \left[\vec{h}_t, \overleftarrow{h}_t\right]$$

(8)

The final output of the Bi-LSTM is the concatenated hidden state sequence for each time step $\overleftarrow{h}_t$, as shown in Eq. (9):

$$H = [h_1, h_2, h_3, \cdots\cdots h_n]$$

(9)

(4) Local Semantic Extraction Based on TextCNN

The MD&A text vectors X output by the Longformer are fed as inputs to the TextCNN, where one-dimensional convolutional filters with window sizes of 2, 3, and 4 are applied to the matrix. Assuming that the input vector size is $n \times h$ and that the filter sizes are $2 \times h$, $3 \times h$, and $4 \times h$, the convolution operation can be represented as Eq. (10):

$$c_i = f(W_i \cdot X + b_i)$$

(10)

where $W_i$ is the weight matrix of the convolution kernel, $b_i$ is the bias term, and the activation function is $f(\cdot)$:

After the convolution, a max pooling operation is applied to select the maximum value from each feature map, preserving the most important features. The max pooling operation for each feature map $c_i$ can be represented as Eq. (11):

$$\hat{c}_i = max\left(c_i\right)$$

(11)

After pooling, the features from different granularities are concatenated to form the final local semantic feature representation of the MD&A text:

$$\hat{C} = [\hat{c}_1, \hat{c}_2, \hat{c}_3]$$

(12)

## Financial and text feature fusion model

For multiperiod MD&A long text semantic extraction, Bi-LSTM is used to extract global semantics, and TextCNN is used to extract local semantics. The two types of semantic features are concatenated to enrich the feature representation and enhance the model's adaptability to complex texts. A fully connected neural network is subsequently used for feature fusion to obtain the final MD&A long text semantic features, which are expected to have better predictive power. Finally,

the extracted MD&A text semantic features are concatenated with the multiperiod financial features and fused via a neural network, as expressed in Eq. (13) and Eq. (14):

$$out = concat\left(out_{final}, \hat{C}\right)$$

(13)

$$z = Wout + b$$

(14)

where out is the concatenated vector of MD&A text semantic features and multiperiod financial features and where $W$ and $b$ are the weight matrix and bias vector, respectively.

This step not only fuses the semantic features with the financial features but also integrates the multiperiod dynamic features of both, enhancing the predictive capability of the model.

## Experiments and results

### Experimental environment and dataset processing

**Experiment environment.** In the experiment, Python was used as the programming language, and the Anaconda environment was utilized for development and management. Anaconda provides powerful package management and virtual environment functionality, making it easier and more efficient to configure the experimental environment, particularly when handling tasks related to deep learning and natural language processing.

**Dataset construction.** The experimental dataset consists of corporate financial indicator data, annual report text data, and financial distress labels. The China Stock Market & Accounting Research (CSMAR) dataset is a widely used database in the field of economic and financial research in China [33,34], and it is highly representative. It covers a wide range of data from the financial and economic sectors, including financial statements of listed companies, market trading data, corporate governance structures, and other related information. Therefore, all experimental data are sourced from the CSMAR database. Companies labeled "ST" (special treatment) are considered to be in financial distress, with the year in which a company is marked as "ST" being identified as the year of financial distress. The corresponding year's dummy variable is set to 1 if the company is in distress and 0 otherwise. For each fiscal year T, the financial distress label is constructed by selecting the data from the corresponding T-2, T-3, and T-4 fiscal years.

This study uses data from all A-share listed companies in China, excluding those in the financial sector, for the years 2016–2023. The financial data from 2012--2021 and the MD&A text data from the companies' annual reports are selected as the feature dataset. Since the number of companies experiencing financial distress with multiperiod historical data is limited, this results in an imbalanced dataset. To address this, stratified undersampling was performed on the majority class samples, selecting companies from the same industry with similar asset sizes as the normal companies to match the distressed companies at a 1:2 ratio. The final dataset consists of 1,359 samples, including 453 positive samples (ST companies) and 906 negative samples (normal companies). In terms of dataset splitting, we conducted multiple rounds of training and evaluation using different proportions. After comprehensive consideration, we ultimately chose a 60:20:20 split ratio, with 60% of the data used as the training set, 20% as the validation set, and 20% as the test set. This ratio strikes a good balance between training efficiency and generalization ability, resulting in relatively stable and favorable performance on both the validation and test sets.

In terms of feature selection, According to existing research [35,36], financial indicators were selected from eight different dimensions: ratio structure, solvency, development capacity, risk level, operational efficiency, per-share indicators, cash flow analysis, and relative value indicators. The MD&A text from the annual reports was used as the textual data. The specific indicators are listed in Table 1.

**Evaluation metrics.** To evaluate the prediction performance of the MTF-FDNN model and other comparison models for corporate financial distress, a confusion matrix was introduced. In the confusion matrix, true positive (TP) represents

**Table 1. Financial and textual indicators.**

| Indicator Type | Indicator Name | Indicator Type | Indicator Name |
|---|---|---|---|
| Solvency | Current Ratio | Per Share Indicators | Earnings per Share |
| | Quick Ratio | | Debt per Share |
| | Cash Ratio | | Earnings per Share |
| | Working Capital/Debt | Relative Value Indicators | Price-to-Earnings Ratio (P/E) |
| | Working Capital | | Price-to-Book Ratio (P/B) |
| | Cash Flow/Current Liabilities Ratio | | Common Stock Profitability |
| | Debt-to-Assets Ratio | Operational Capability | Receivables-to-Revenue Ratio |
| | Long-Term Debt/Total Assets | | Accounts Receivable Turnover |
| | Tangible Assets Debt Ratio | | Inventory-to-Revenue Ratio |
| | Long-Term Capital Debt Ratio | | Inventory Turnover Ratio |
| | Long-Term Debt/Working Capital | | Accounts Payable Turnover |
| | Long-Term Debt to Equity Ratio | | Working Capital (Capital) Turnover |
| Ratio Structure | Current Asset Ratio | | Total Asset Turnover |
| | Cash Asset Ratio | | Equity Turnover Ratio |
| | Fixed Asset Ratio | Development Ability | Capital Accumulation Rate |
| | Current Liability Ratio | | Total Asset Growth Rate |
| | Operational Liability Ratio | | Basic Earnings per Share Growth Rate |
| Cash Flow Analysis | Net Profit Cash Net Content | | Net Profit Growth Rate |
| | Operating Income Cash Content | | Operating Profit Growth Rate |
| | Operating Profit Cash Net Content | | Operating Income Growth Rate |
| | Operating Index | | Owner's Equity Growth Rate |
| Risk Level | Financial Leverage | | Net Asset per Share Growth Rate |
| | Operating Leverage | Text | MD&A Text |

the number of companies that are actually in financial distress and are correctly predicted by the model as being in distress, i.e., the number of companies the model successfully identifies as distressed. False positive (FP) represents the number of companies that are actually not in financial distress but are incorrectly predicted by the model as being in distress. A false negative (FN) represents the number of companies that are actually in financial distress but are incorrectly predicted by the model as not being in distress. True negative (TN) represents the number of companies that are actually not in financial distress and are correctly predicted by the model as not being in distress.

Four metrics were selected for a comprehensive evaluation: the area under the ROC curve (AUC), accuracy, recall, and F1 score. The specific formulas for calculation are as follows:

**AUC**

$$TPR = \frac{TP}{TP+FN} \tag{15}$$

$$FPR = \frac{FP}{TN+FP} \tag{16}$$

The ROC curve is plotted with the true positive rate (TPR) on the vertical axis and the false positive rate (FPR) on the horizontal axis. The AUC is the area under the ROC curve. The closer the AUC value is to 1, the better the performance of the model.

## Accuracy

The accuracy in Eq. (17) represents the proportion of correctly predicted companies out of the total number of companies. A higher accuracy indicates that the model is more accurate in predicting the overall financial status of companies.

$$Accuracy = \frac{TP+TN}{TP+FP+TN+FN} \qquad (17)$$

## Recall

Eq. (18) represents the proportion of companies that are correctly identified as being in financial distress by the model, i.e., the percentage of companies that are actually in financial distress and are correctly predicted as "financial distress" by the model. A higher recall indicates that the model is able to identify most of the companies in financial distress.

$$Recall = \frac{TP}{TP+FN} \qquad (18)$$

## F1 score

The F1 score in Eq. (19) is the harmonic mean of precision and recall. A higher F1 score indicates that the model performs well in balancing the detection of distressed companies and avoiding false positives.

$$F1-score == \frac{2\,Precision * Recall}{Precision+Recall} \qquad (19)$$

In conclusion, the closer the values of these metrics are to 1, the better the model's prediction performance.

## Comparison methods

To validate the effectiveness of multiperiod annual report text in financial distress prediction and the superiority of the proposed model, the predictive performance of the MTF-FDNN model is compared with that of other financial distress prediction models. The comparison models are as follows:

### Financial distress prediction model based on traditional machine learning methods

These models predict financial distress by integrating multiperiod financial indicators and company annual report text information. Multiperiod financial indicators are unfolded into flat data, and TF-IDF is used to extract multiperiod text features. The two types of features are then fused to form the input for the model. Traditional machine learning algorithms such as logistic regression (LR), decision tree (DT), random forest (RF), and extreme gradient boosting (XGBoost) are used for model training.

### Financial distress prediction model based on multi-period text data encoded by BERT

This model first filters and truncates the annual report text to retain the most valuable content. Then, it applies a pretrained BERT model to encode the text data, generating high-dimensional semantic vectors that capture the implicit contextual relationships and deep patterns within the text. On the basis of these vectors, the model further integrates the sequential feature extraction capability of Bi-LSTM and the local key feature extraction ability of TextCNN to effectively fuse multiperiod text information.

### Single-period annual report text fusion deep neural network model (BiLSTM&TextCNN)

This model combines company annual report text data and financial data for prediction. It extracts the text information from a company's single annual report and processes and extracts the semantic features from the report via the fusion

deep neural network of Bi-LSTM and TextCNN. The model focuses on the Management Discussion and Analysis (MD&A) section of the annual report. This text feature extraction model is capable of capturing the implicit trends and risk signals of the company within the text.

### Multi-period annual report text fusion deep neural network model (BiLSTM&TextCNN)

This model is a deep learning model that analyzes company annual report text data across multiple years. It extracts the semantic features from annual reports of multiple years and processes and integrates the text from different periods via Bi-LSTM and TextCNN. The model captures long-term trends and potential risks in annual reports. By analyzing time series text data, the model better reflects a company's evolving business conditions over time, which helps improve the comprehensiveness and accuracy of financial distress prediction.

### Parameter sensitivity analysis

To evaluate the MTF-FDNN model's sensitivity to key hyperparameters, this study systematically adjusted and tested hidden_size_t, num_layers, dropout, learning rate, batch_size, and concat_size, analyzing their impact on model performance. The results are as follows:

### Bi-LSTM hidden layer dimension

In the sensitivity analysis of the Bi-LSTM hidden layer dimension (hidden_size_t), as shown in Fig. 3(a), the model performance exhibited a rising-then-falling trend as the dimension increased from 2. The best performance was achieved at a

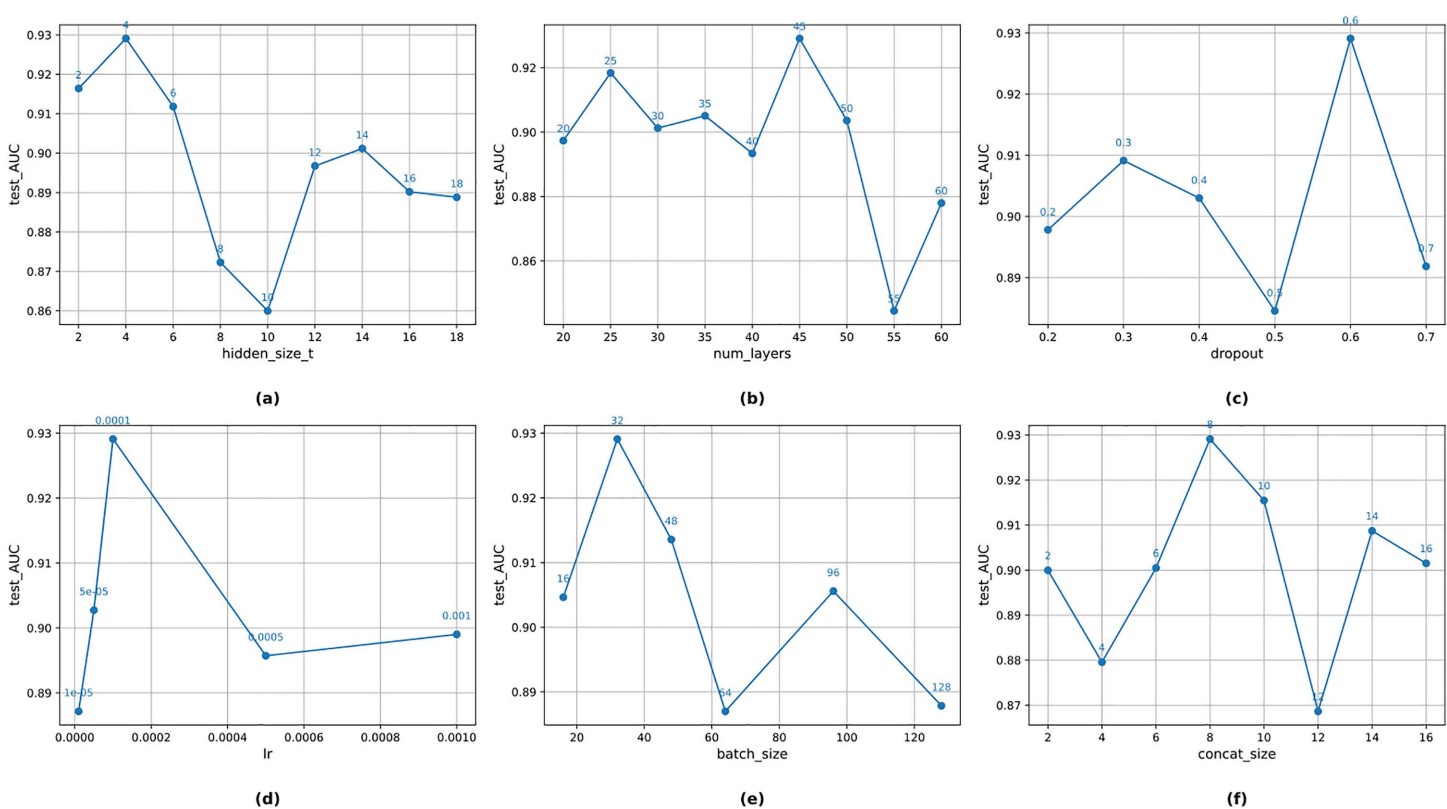

**Fig 3. Parameter Sensitivity Analysis.**

dimension of 4, indicating an optimal balance between feature abstraction capability and model complexity. Increasing the dimension to 6 and 8 led to performance degradation, likely due to redundant parameters increasing training difficulty and causing overfitting. When the dimension reached 10 or higher, performance continued to decline, suggesting that excessive capacity negatively affected generalization. Therefore, a dimension of 4 was selected as the optimal configuration.

### Number of Bi-LSTM layers

As shown in Fig. 3(b), the number of LSTM layers (num_layers) had a significant impact on model performance. Increasing the number of layers from 20 to 45 resulted in continuous performance improvement, indicating that moderately deepening the network enhances the abstraction and representation of temporal features. However, when the number of layers exceeded 45, performance declined, possibly due to gradient vanishing, unstable training, and increased overfitting risk. Considering both performance and training stability, 45 layers were determined to be the optimal depth.

### Dropout ratio

In the sensitivity analysis of the dropout ratio, as shown in Fig. 3(c), ratios between 0.2 and 0.4 maintained good representation capability but lacked sufficient regularization, leading to higher overfitting risk. When the ratio was increased to 0.6, the model's generalization ability was significantly enhanced, achieving peak performance. At a ratio of 0.7, although regularization was stronger, excessive feature dropout impaired learning ability, thereby reducing performance. Thus, 0.6 was selected as the optimal dropout ratio.

### Learning rate

As shown in Fig. 3(d), the learning rate had an important influence on both training stability and performance. A lower learning rate ensured stability but slowed convergence, limiting final performance. When set to 1e-4, the model maintained stability while converging at a reasonable speed, achieving the best performance. Increasing the learning rate to 5e-4 or 1e-3 accelerated training but reduced performance, likely because the step size was too large, causing the optimizer to overshoot the optimal solution. Therefore, 1e-4 was selected as the optimal learning rate.

### Training batch size

In terms of batch size, a size of 16 yielded relatively good performance, indicating that small batches can enhance generalization to some extent. Increasing the batch size to 32 and 48 further improved performance, suggesting that a moderate batch size can balance training efficiency and gradient estimation accuracy. However, when the batch size exceeded 64, performance dropped, possibly due to reduced gradient update frequency and increased GPU memory usage. Consequently, a batch size of 32 was chosen to balance performance and computational resource utilization.

### Feature concatenation dimension

In the sensitivity analysis of the feature concatenation dimension (concat_size), as shown in Fig. 3(f), a smaller concatenation dimension maintained basic representation capability but limited the integration of multi-source features. Increasing the dimension to 8 and 10 significantly improved performance, indicating that moderate enlargement enhances feature richness. However, when exceeding 12, performance fluctuated or declined, likely due to redundant features and higher computational burden. Thus, a dimension of 8 was selected as optimal.

Based on the above analysis, the model achieves an optimal balance between performance and computational resource consumption with the parameter configuration shown in **Table 2**. This parameter set not only performs excellently in individual sensitivity tests but also demonstrates good stability and generalization capability throughout the overall training process, laying a solid foundation for subsequent model optimization and practical applications.

**Table 2. Arameter Settings.**

| Hyperparameter | Description | Optimal Value |
|---|---|---|
| hidden_size_t | Bi-LSTM hidden layer dimension | 4 |
| num_layers | Number of Bi-LSTM layers | 45 |
| dropout | Dropout ratio | 0.6 |
| Learning rate | Learning rate | 1e-4 |
| batch_size | Training batch size | 32 |
| concat_size | Feature concatenation dimension | 8 |

## Comparison experiments and result analysis

### Comparison of results between traditional machine learning methods and multi-period annual report financial and text data

To validate the advantage of the proposed MTF-FDNN fusion deep neural network over traditional machine learning models in financial distress prediction, this study first inputs the unfolded multiperiod indicators and TF-IDF features of multi-period text into machine learning models for prediction. The prediction results from these models are then compared and analyzed against the prediction results of the MTF-FDNN model.

The experimental results are shown in Table 3 and Fig 4, the MTF-FDNN model outperforms traditional machine learning models across all the metrics, demonstrating significant advantages. Specifically, the AUC of the MTF-FDNN reaches 0.9291, which is higher than those of the other models, indicating stronger discriminative power. The accuracy (ACC) is 0.8600, surpassing those of random forest and XGBoost. Notably, in terms of recall, the MTF-FDNN achieves 0.8711, significantly outperforming the random forest, indicating its stronger ability to identify financially distressed companies. Furthermore, its F1 score is 0.8531, yielding the best overall performance. In contrast, traditional models have limited capability in handling complex features and cannot match the performance of the MTF-FDNN.

According to the T-test results in Table 4, MTF-FDNN sws a significant advantage in financial distress prediction. Compared to traditional models such as LR and DT, MTF-FDNN achieves a notable improvement in AUC and accuracy, demonstrating its superior ability to distinguish distressed companies. Most importantly, MTF-FDNN significantly outperforms all other models in recall and F1 score, making it more effective at detecting high-risk enterprises, which is crucial in financial risk prediction.

### Comparison of financial distress prediction results using multi-period text data with different encoding methods

To verify the performance of Longformer in processing long texts, this study uses both Longformer and BERT for text encoding and inputs them into the MTF-FDNN model. The results in Table 5 and Fig 5 shows that Longformer outperforms BERT. In terms of the AUC, Longformer reaches 0.9291, which is higher than that of BERT. In terms of accuracy, Longformer's 0.8566 significantly outperforms BERT, indicating stronger classification capability. In terms of recall, Longformer outperforms BERT with values of.8711 and 0.80620, respectively. However, Longformer achieves an F1 score of 0.8531, significantly outperforming BERT. This demonstrates that the Longformer is more effective in capturing the deep

**Table 3. Comparison of Results from Traditional Machine Learning Methods.**

| | LR | DT | RF | XGBoost | MTF-FDNN |
|---|---|---|---|---|---|
| AUC | 0.8542 | 0.8104 | 0.8634 | 0.8718 | 0.9291 |
| ACC | 0.7854 | 0.7956 | 0.8146 | 0.8103 | 0.8600 |
| Recall | 0.7703 | 0.7368 | 0.7506 | 0.7305 | 0.8711 |
| F1 | 0.6875 | 0.7643 | 0.7468 | 0.6855 | 0.8531 |

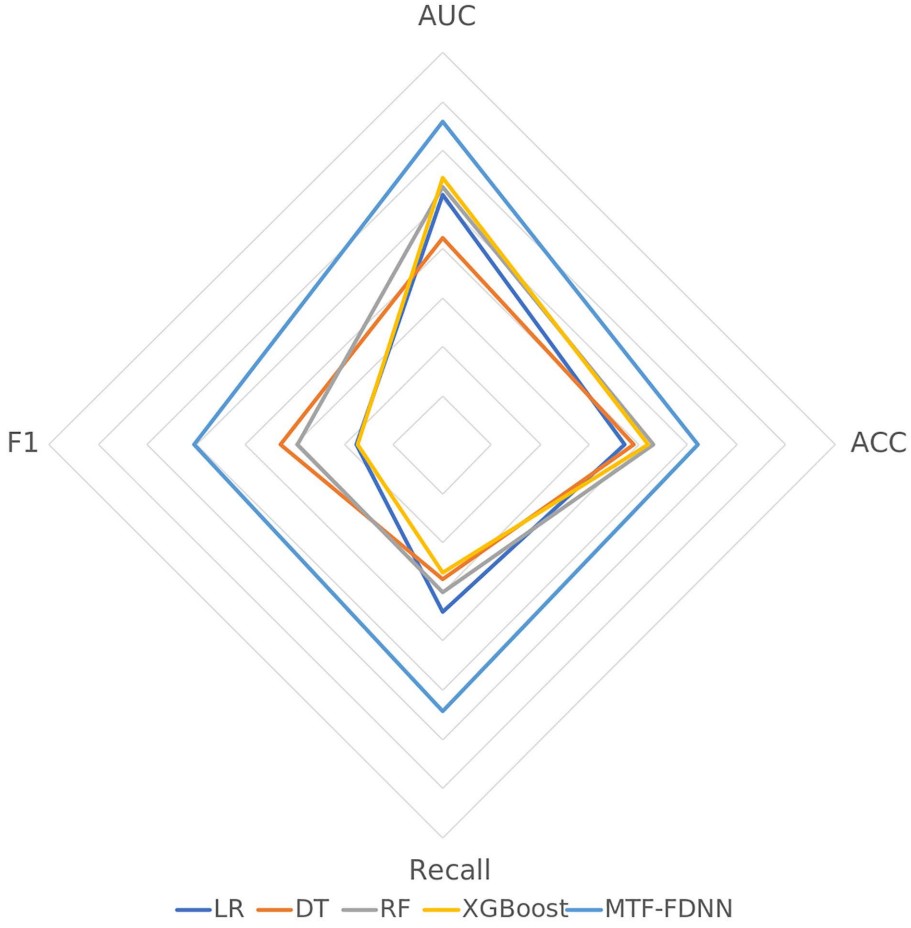

**Fig 4. Radar Chart of the Comparison of the Traditional Machine Learning Method Results.**

**Table 4. T-test results(Values in bold indicate significant differences at 90% confidence).**

|  | MTF-FDNN &LR | MTF-FDNN &DT | MTF-FDNN &RF | MTF-FDNN &XGBoost |
|---|---|---|---|---|
| AUC | **0.0232** | **0.0008** | **0.0777** | 0.1312 |
| ACC | **0.0851** | **0.0610** | 0.2359 | **0.0675** |
| Recall | **0.0024** | **0.0013** | **0.0024** | **0.0004** |
| F1 | **0.0633** | **0.0142** | **0.0152** | **0.0251** |

semantic features of long texts, providing better input for the MTF-FDNN model, thereby enhancing its overall predictive performance. Finally, a significance test was conducted, and the t-test results show that the Longformer encoding method significantly outperforms BERT in all indexes.

## Financial distress prediction model based on multi-period text data

To verify the incremental effect of multiperiod annual report text data over single-period text data, this study first inputs single-period and multiperiod annual report texts into Bi-LSTM+TextCNN for comparison.

**Table 5. Comparison of Text Feature Extraction between Longformer and BERT.**

|  | MTF-FDNN(BERT) | MTF-FDNN(Longformer) |
| --- | --- | --- |
| AUC | 0.8776 | 0.9291 |
| ACC | 0.7582 | 0.8600 |
| Recall | 0.8062 | 0.8711 |
| F1 | 0.7291 | 0.8531 |

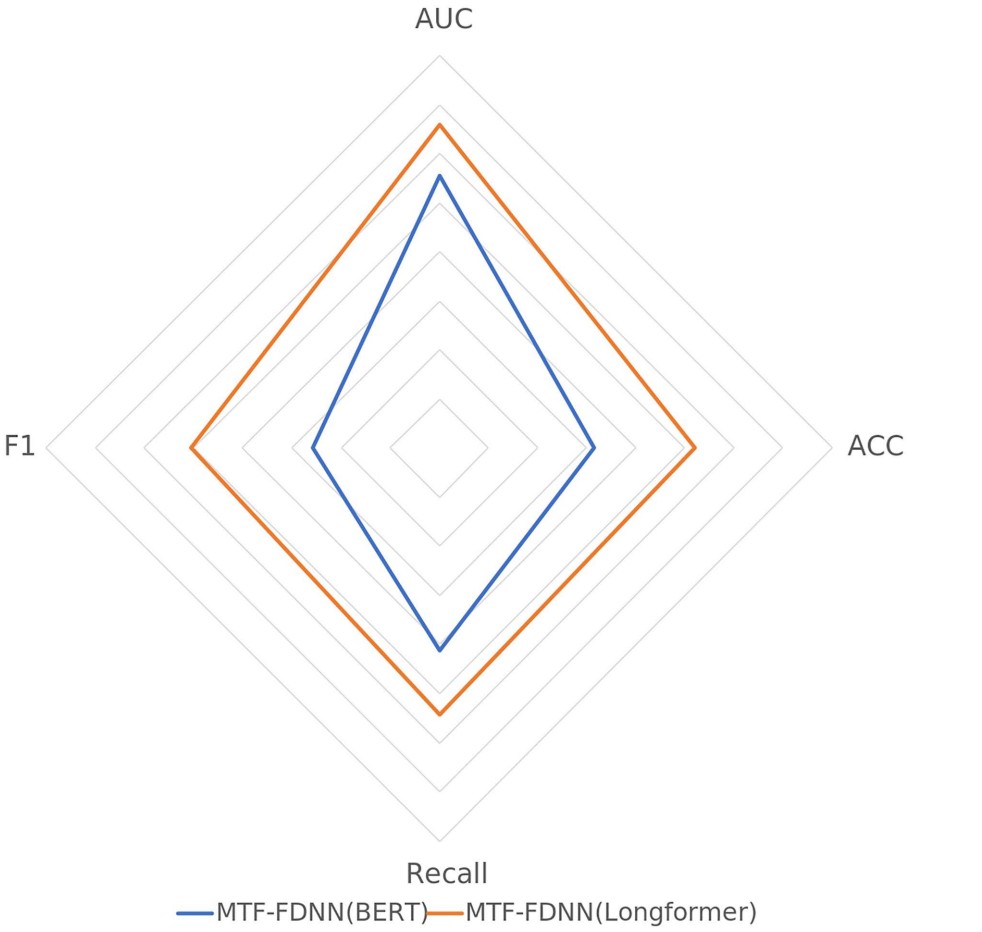

**Fig 5. Radar Chart Comparing Longformer and BERT Text Feature Extraction.**

According to the results shown in Table 6 and Fig 6, the prediction performance on the basis of multiperiod annual report text data surpasses that of single-period text data across all evaluation metrics. Compared with that of single-period annual reports, the AUC value of multiperiod annual reports increased by 6.57%, indicating better performance in distinguishing between positive and negative samples. The model's accuracy improved by 4.73%, demonstrating that multiperiod annual report text data provide more precise overall predictions. The recall rate increased by 3.52%, reflecting an enhanced ability to identify financially distressed companies. The F1 score increased by 2.51%, indicating a better balance between precision and recall. Finally, a significance test was conducted, and the t-test results indicate that incorporating multi-period text data for prediction significantly enhances the AUC, accuracy, and recall compared to using

**Table 6. Prediction performance of single-period and multiperiod annual report texts.**

|  | Single-period Text | Multiperiod Text | Information Gain |
|---|---|---|---|
| AUC | 0.8062 | 0.8719 | 6.57% |
| ACC | 0.7582 | 0.8055 | 4.73% |
| Recall | 0.7042 | 0.7394 | 3.52% |
| F1 | 0.7177 | 0.7428 | 2.51% |

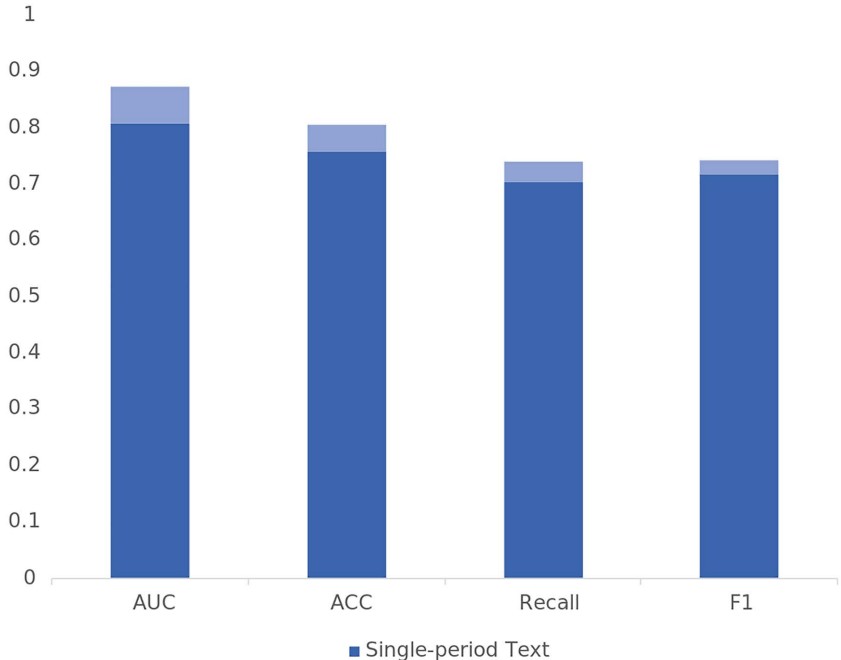

**Fig 6. Prediction performance of single-period and multiperiod annual report texts.**

single-period text data. In conclusion, by integrating multiperiod annual report text data, the model's predictive power is significantly strengthened, with all evaluation metrics showing notable improvement.

## Ablation experiments and results analysis

### Comparison of single models based on multi-period annual report text data

To further validate the effectiveness of the model design, this study inputs the multiperiod annual report text data encoded by Longformer into individual BiLSTM and TextCNN models for modeling and compares the results with those of the fused BiLSTM and TextCNN model.

The experimental results in Table 7 and Fig 7 indicate that the BiLSTM and TextCNN fusion model for text feature extraction in the MTF-FDNN model outperforms the single feature extraction models across all the metrics, demonstrating the advantages of the fusion feature extraction method. In terms of the AUC metric, the BiLSTM&TextCNN model achieves an AUC of 0.8719, which is significantly higher than that of both single models, indicating superior discrimination ability. In terms of accuracy, the BiLSTM&TextCNN model achieves an accuracy of 0.8055, which is notably higher than that of both individual models and thus has a stronger classification ability. Additionally, the BiLSTM&TextCNN model has

**Table 7. Comparison of the prediction results of the BiLSTM and TextCNN single models.**

|  | TextCNN | BiLSTM | BiLSTM&TextCNN |
|---|---|---|---|
| AUC | 0.8072 | 0.8054 | 0.8719 |
| ACC | 0.7291 | 0.7255 | 0.8055 |
| Recall | 0.7239 | 0.7455 | 0.7394 |
| F1 | 0.7347 | 0.7236 | 0.7428 |

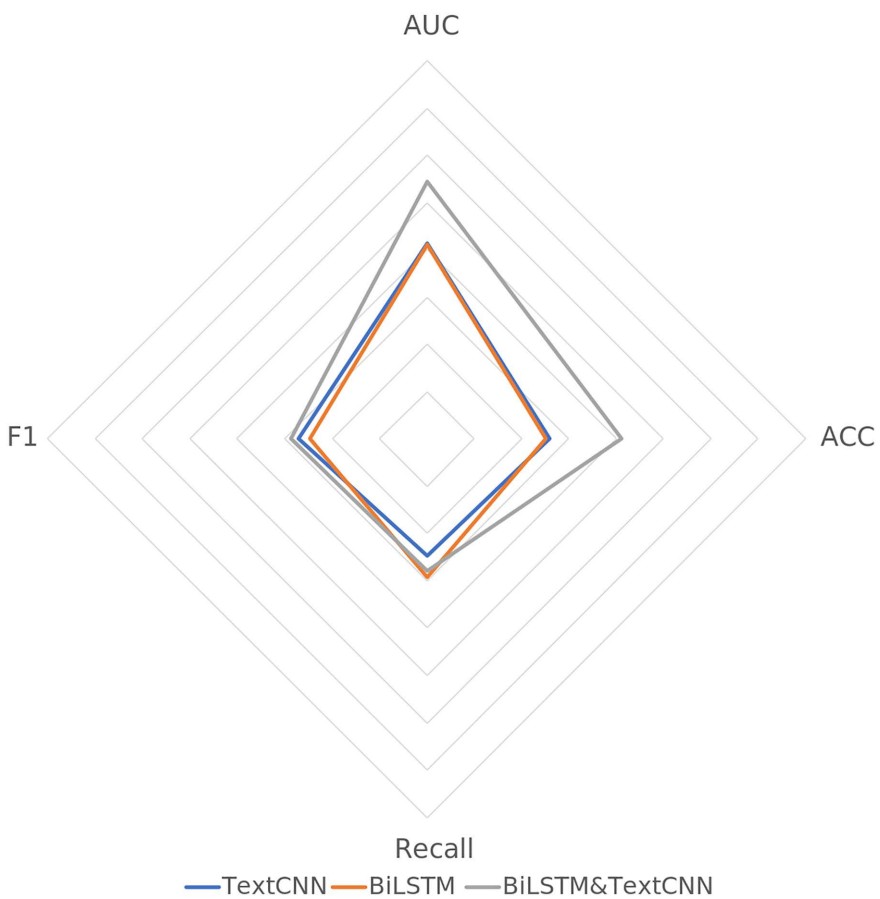

**Fig 7. Prediction Results of the BiLSTM and TextCNN Single Models – Radar Chart.**

a recall rate of 0.7394, which is higher than those of both BiLSTM and TextCNN, suggesting a more comprehensive identification of positive samples. Furthermore, the F1 score of the BiLSTM&TextCNN model is 0.7428. The t-test results show that, compared to a single model, the ensemble of BiLSTM and TextCNN significantly improves the AUC and accuracy, which is a notable improvement over the single model, further validating the effectiveness of the design that integrates the BiLSTM and TextCNN feature extraction modules.

## Financial distress prediction results based on multi-period annual report text and financial data

To validate the information gain brought by multiperiod financial data and multiperiod text data, this study compares the MTF-FDNN model, which integrates multiperiod annual report text and financial data, with the following models: the

BiLSTM +TextCNN model, which is based on multiperiod annual report text data, and the fully connected neural network model, which is based on multiperiod financial data.

According to the results shown in Table 8 and Fig 8, after incorporating multiperiod financial data, the AUC, accuracy, recall, and F1 score improved by 4.98%, 6.54%, 10.58%, and 6.48%, respectively. Among these, recall showed the largest improvement, indicating a significant enhancement in the model's ability to identify financial distress samples after adding multiperiod financial indicators.

After incorporating multiperiod text data, the performance of the MTF-FDNN model also significantly improved. Specifically, the AUC increased by 5.72%, demonstrating more precise classification of positive and negative samples; accuracy improved by 5.45%, suggesting that the model's overall prediction reliability increased; recall increased by 13.17%, reflecting a reduced likelihood of false negatives in identifying financial distress companies; and the F1 score increased by 11.03%, indicating that the model became more robust in handling imbalanced data. Finally, the significance test shows that the prediction results combining financial indicators and textual information significantly outperform the predictions using only multi-period text data or only multi-period financial indicators.

## Conclusion and future works

This study proposes a fusion deep neural network model for financial distress prediction (MTF-FDNN) based on multi-period annual report texts and financial data. The model innovatively combines multi-period financial features with long-text semantic features, aiming to improve the accuracy of corporate financial distress prediction. In the model design, we constructed

**Table 8. Information Gain Effects of Multi-Period Text and Multi-Period Financial Data.**

|  | Multiperiod Text Data | Multiperiod Financial Data | Multiperiod Financial Data＋Multiperiod Text Data |
|---|---|---|---|
| AUC | 0.8719 | 0.8793 | 0.9291 |
| ACC | 0.8055 | 0.7946 | 0.8600 |
| Recall | 0.7394 | 0.7653 | 0.8711 |
| F1 | 0.7428 | 0.7883 | 0.8531 |

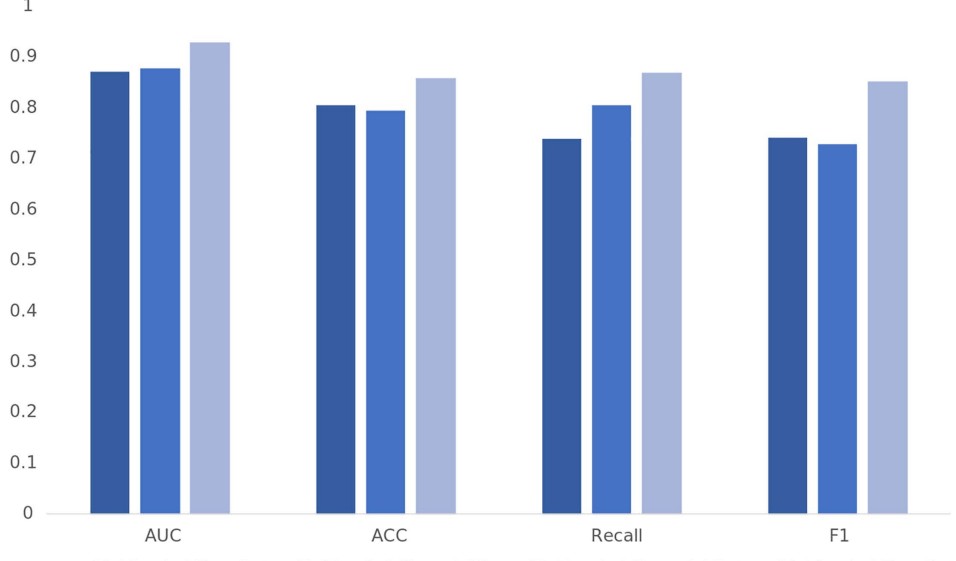

**Fig 8. Information Gain Effect of Multi-Period Text and Multi-Period Financial Data.**

a fully connected neural network (FNN) as the financial feature extraction module and combined fine-tuned Longformer, Bi-LSTM, and TextCNN for text feature extraction. Longformer employs a sliding window attention mechanism, enabling it to handle ultra-long texts, making it particularly suitable for analyzing the Management Discussion and Analysis (MD&A) section in annual reports, capturing global information in the text. Meanwhile, Bi-LSTM and TextCNN strengthen the modeling capabilities of temporal dependencies and local features, respectively. Through this multi-layered and multi-modal feature fusion, MTF-FDNN effectively mines latent information in annual reports and enhances the model's predictive capability.

Parameter sensitivity analysis was conducted to systematically evaluate the effects of six key hyperparameters—hidden_size, num_layers, dropout, learning rate, batch_size, and concat_size—on model performance, aiming to achieve an optimal balance between predictive accuracy and computational cost. The optimal configuration was determined as *hidden_size_t = 4*, *num_layers = 45*, *dropout = 0.6*, *learning rate = 1e-4*, *batch_size = 32*, and *concat_size = 8*. This combination not only achieved strong performance in individual sensitivity tests but also demonstrated excellent stability and generalization capability throughout the entire training process, providing a solid parameter foundation for the high performance of MTF-FDNN in financial distress prediction.

When compared with traditional machine learning models, such as logistic regression (LR) and decision trees (DT), which have advantages in training speed and interpretability, their predictive accuracy is relatively low due to their inability to capture complex nonlinear relationships. Random Forest (RF) and XGBoost improve prediction accuracy by integrating multiple weak classifiers, but their performance is still limited by their ability to process long texts. In contrast, MTF-FDNN, by combining financial data and annual report texts, can more comprehensively capture both the financial performance and potential risks of enterprises, particularly excelling in handling long texts. Experimental results show that MTF-FDNN significantly outperforms these traditional machine learning models in key metrics such as AUC, F1-score, and recall, further validating its effectiveness in financial distress prediction. The superiority of the model was further verified through comparisons of different encoding methods. Although models based on BERT perform well in text semantics, the sliding window mechanism of Longformer shows superior performance in handling ultra-long texts, surpassing the standard BERT model in semantic capture. In ablation experiments, single TextCNN or Bi-LSTM models, relying only on local convolutional or global features, struggled to process the complex semantic information in long texts. However, the hybrid architecture combining TextCNN and Bi-LSTM, through the fusion of global and local information, captured the multi-dimensional information in both financial data and annual report texts more stably and accurately. Furthermore, by comparing the incremental effect of multi-period annual report text data versus single-period text data on model performance, we confirmed the significant information gain brought by multi-period annual report text data. Finally, the combination of multi-period financial data and multi-period annual report text data showed a gain effect, providing the model with richer and more comprehensive information, thereby enhancing the accuracy of financial distress prediction. This finding further confirms the superiority of MTF-FDNN in processing multi-period data.

Moreover, compared to traditional models, MTF-FDNN offers a more reasonable trade-off between computational efficiency and accuracy. First, in terms of computational efficiency, MTF-FDNN significantly improves the efficiency of handling long texts by replacing the standard Transformer architecture with Longformer. Longformer optimizes the self-attention computation using a sliding window attention mechanism, reducing computational complexity when processing ultra-long texts, such as the MD&A sections in corporate annual reports, compared to traditional models like BERT. This greatly reduces the computational overhead, thus enhancing both training and inference efficiency. Second, in terms of accuracy, MTF-FDNN, by combining multi-period financial data and annual report text information, can comprehensively consider the potential impact of both financial performance and textual content, improving the accuracy of financial distress prediction. In contrast, traditional machine learning models (e.g., LR, DT, RF, and XGBoost) mainly rely on structured data and fail to effectively capture unstructured information from texts. MTF-FDNN, by combining Longformer, Bi-LSTM, and TextCNN, extracts both global and local information from the text, significantly improving prediction accuracy.

Although MTF-FDNN demonstrates significant advantages in financial distress prediction, this study still has limitations. First, the model's hyperparameter optimization is still in its early stages, with room for improvement in fine-tuning

learning rate schedules and regularization parameters, such as adaptive learning rate decay and weight decay coefficient optimization. Second, the data input dimensions are relatively narrow, currently relying solely on annual report text data, and have not integrated multi-source heterogeneous data such as ESG reports, news sentiment, and social media, which limits the model's ability to capture the multidimensional risks of enterprises. Third, the method for addressing class imbalance (undersampling the majority class) may result in the loss of minority class information, and the potential of current data augmentation techniques (e.g., EDA, Back-Translation) and generative adversarial networks (GANs) has not been fully explored. Future research could expand in three directions:

(1) introducing hierarchical attention mechanisms (such as Hierarchical Attention Networks) to build a "word-sentence-paragraph" three-level semantic representation, thereby enhancing the long-text context modeling ability;

(2) developing a multi-modal data fusion framework that integrates unstructured text (annual reports/news) with structured data (financial indicators/macroeconomic variables) to form a comprehensive risk assessment system;

(3) for imbalanced data, combining data generation techniques (e.g., Conditional GAN) with algorithmic improvements (e.g., focal loss function, ensemble learning strategies) to construct a more robust predictive model.

These improvements will promote systematic advancements in model generalization, interpretability, and data utilization efficiency, providing more reliable methodological support for financial risk warning.

Finally, the findings of this study have broad application potential in the financial field, especially in credit risk assessment, corporate bankruptcy prediction, and audit risk management. The model can provide more accurate financial distress prediction tools for financial institutions, investors, and regulatory bodies. More importantly, MTF-FDNN is not limited to a single industry or region, and its cross-industry and cross-regional adaptability will further validate its generalization ability. Additionally, future research can further optimize text data processing methods, explore multi-modal fusion technologies, and improve the model's interpretability and computational efficiency. With continuous model optimization and application, financial distress prediction will become more precise and stable, providing scientific support for more decision-making processes and bringing profound impact to the financial industry.

## Author contributions

**Data curation:** Zhiyi Wang.

**Formal analysis:** Jiawang Li.

**Funding acquisition:** Chongren Wang.

**Methodology:** Jiawang Li.

**Project administration:** Pimei Gong.

**Software:** Pimei Gong.

**Supervision:** Chongren Wang.

**Validation:** Pimei Gong.

**Visualization:** Zhiyi Wang.

**Writing – original draft:** Pimei Gong.

**Writing – review & editing:** Chongren Wang.

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
