## [Decision Letter · Decision Letter 0]

24 Feb 2025

PONE-D-25-01170Corporate financial distress prediction with multiperiod annual report data: A fusion deep neural network modelPLOS ONE

Dear Dr. Gong,

Thank you for submitting your manuscript to PLOS ONE. After careful consideration, we feel that it has merit but does not fully meet PLOS ONE’s publication criteria as it currently stands. Therefore, we invite you to submit a revised version of the manuscript that addresses the points raised during the review process.

We look forward to receiving your revised manuscript.

Kind regards,

Jiwei Tian

Academic Editor

PLOS ONE

Journal Requirements:

“Shandong Province Science and Technology based Small and Medium sized Enterprises Innovation Capability Enhancement Project (2023TSGC0208)

Jinan City School Integration Development Strategy Project (JNSX2023052).”

Reviewers' comments:

Reviewer's Responses to Questions

**Comments to the Author**

1. Is the manuscript technically sound, and do the data support the conclusions?

Reviewer #1: Yes

Reviewer #2: Partly

2. Has the statistical analysis been performed appropriately and rigorously? 

Reviewer #1: Yes

Reviewer #2: Yes

3. Have the authors made all data underlying the findings in their manuscript fully available?

Reviewer #1: Yes

Reviewer #2: Yes

4. Is the manuscript presented in an intelligible fashion and written in standard English?

Reviewer #1: Yes

Reviewer #2: Yes

5. Review Comments to the Author

Reviewer #1: The paper proposes a Fusion Deep Neural Network model based on Multiple annual Reports Text data and Financial data (MTF-FDNN). This model is capable of extracting long-text features from multi-period corporate annual reports as well as features from multi-period financial indicators. To further improve the manuscript, the following suggestions are given:

1、In the paper, some figures in the manuscript are a little blurry, please improve the clarity.

2、The formatting specifications throughout the manuscript need to be carefully checked and revised. For example, formulas need to be centered.

3、Since there are some papers in this topic, the contributions of the manuscript should be better summarized and listed.

4、While the introduction sets the context, a more explicit literature review section could better situate the study within the broader research landscape, such as Financial distress prediction with annual reports-based deep textual feature extraction: A hybrid approach, EVADE Targeted Adversarial False Data Injection Attacks for State Estimation in Smart Grid, and so on. These references could provide valuable insights into your research.

5、Add a section on the limitations of the work and future work in this paper.

Reviewer #2: The introduction section

1. The introduction could be strengthened by incorporating more recent literature to better contextualize the study. For example, the sentence: “Previous studies have shown that predictive models that combine textual and financial data can significantly increase the accuracy of financial distress predictions.” lacks citations for the referenced studies. Similarly, the authors state: “In the early stages of financial distress prediction research, scholars domestically and internationally relied primarily on financial indicators from a single fiscal year, applying machine learning or deep learning techniques for prediction. In recent years, text mining has become a hot topic in academia.” Again, no supporting references are provided. Including relevant citations would enhance the credibility of these claims.

2. A clearer transition from the problem statement to the proposed solution would improve the overall flow of the introduction.

3. The choice of Longformer, Bi-LSTM, and TextCNN requires stronger theoretical justification. Why is Longformer better than other transformers? Why were Bi-LSTM and TextCNN combined instead of alternative architectures (e.g., hierarchical attention networks)? Providing a rationale for these choices would strengthen the methodological foundation of the study.

4. Adding a brief description of the structure of the article at the end of the introduction would enhance its clarity and help guide the reader through the following sections.

The related works section

5. Some of the references cited in the manuscript are relatively old (e.g., Beaver, 1966). Additionally, the manuscript does not contain many sources, and most references are older than five years.

For example, you state “Hajek P. et al. [15] applied XGBoost and AdaBoost algorithms for classification analysis of samples that integrate textual metrics. García V. et al. [16] further improved the classification accuracy of models by combining ensemble strategies such as bagging and random subspace methods including more recent studies would provide a more current context. ..”, the mentioned authors, as well as others, have more recent studies in this subject area. Including newer literature would help provide a more up-to-date context.

6. The organization of related studies into subsections is appreciated, as it enhances the clarity of the article. However, the discussion of individual studies is too brief. The authors introduce the problems each study addresses but do not provide an in-depth discussion of the methodologies or results. A more detailed analysis of prior work, including quantitative performance metrics (e.g., accuracy rates of previous models), would improve the quality of this section. Furthermore, while the limitations of prior studies are mentioned, the research gaps they leave unaddressed are not explicitly stated. Identifying these gaps more clearly would help to better understand the described studies within The related works section.

The experiments and results section

7. The CSMAR dataset is used, but there is no justification for why this dataset is representative. More information about the dataset, such as the number of companies, the time span covered, and other relevant characteristics, should be provided. Additionally, there is no discussion of cross-validation or alternative training/test splits to ensure the model avoids overfitting. The authors should explain why they chose a 60:20:20 split and whether they experimented with other ratios (e.g., 70:20:10).

8. The comparison between traditional machine learning models and MTF-FDNN is clear, but the claims of superiority would be more robust with statistical significance testing (e.g., p-values). Adding such analyses would strengthen the validity of the results.

Conclusion and future works sections

9. The conclusion could be more reflective, discussing the broader impact and potential applications of the proposed model.

10. A deeper comparison between the proposed MTF-FDNN model and previous financial distress prediction models mentioned in the Related works section is missing. While the article presents results, it does not explicitly compare them to previous state-of-the-art models in financial distress prediction. The authors should clarify why MTF-FDNN is superior to BERT-based models, CNN-only models, or other hybrid architectures.

11. A discussion of computational efficiency and trade-offs between accuracy and interpretability would be beneficial. Addressing these aspects would provide a more balanced assessment of the model's practical implications.

12. There is no discussion including the limitations and implications of the research.

Overall, the structure of the research article is non-standard.

6. PLOS authors have the option to publish the peer review history of their article (what does this mean? ). If published, this will include your full peer review and any attached files.

**Do you want your identity to be public for this peer review?** For information about this choice, including consent withdrawal, please see our Privacy Policy .

Reviewer #1: No

Reviewer #2: No

---

## [Author Response · Author response to Decision Letter 1]

3 Apr 2025

Dear Editors:

This is a resubmitted manuscript by Chongren Wang, Pimei Gong, Jiawang Li and Zhiyi Wang, entitled “Corporate financial distress prediction with multiperiod annual report data: A fusion deep neural network model” with Manuscript [PONE-D-25-01170]. We have carefully revised this paper according to the editorial decision letter one by one and explains how the paper has been changed point-to-point. See Section 1 for a detailed description.

The authors have read and approved this version of the article, and due care has been taken to ensure the integrity of the work. No part of this paper has been published or submitted elsewhere. No conflict of interest exits in the submission of this manuscript.

This paper elaborately details a fused deep neural network model based on multiperiod annual report text and financial indicator data to predict corporate financial distress. The main contributions of this study are as follows:

(1) Development of a feature fusion model to extract semantic features from long texts.

This study applies multiperiod financial report texts to the field of financial distress prediction and designs a novel neural network architecture tailored to the characteristics of multiperiod financial report texts. This approach enhances the prediction performance of financial distress models that incorporate long texts from multiperiod financial reports. First, the pretrained Longformer model is utilized to compute word vectors for the long texts in financial reports. Subsequently, the global features of the text were extracted via bidirectional long short-term memory (Bi-LSTM), and the local features were extracted via a Text Convolutional Neural Network (TextCNN). A feature-level fusion method is then used to construct multiperiod semantic features of long texts, offering a new perspective for extracting semantics from multiperiod financial report texts.

(2) A dynamic financial distress prediction model is constructed on the basis of multiperiod annual report data.

A dynamic financial distress prediction model is constructed on the basis of multimodal data from multiperiod annual report texts and financial indicators. First, multiple fully connected neural networks are used to process multiperiod financial indicator data. Next, a fusion model is designed to handle multiperiod annual report texts by extracting long texts from the Management Discussion and Analysis (MD&A) sections of multiple years. The feature fusion model is then used to compute the global and local semantic features of these long texts, and the results are integrated as multiperiod text semantic features. Finally, a neural network combines the extracted multiperiod textual semantics and multiperiod financial features for predictive computation.

We believe the paper may be of particular interest to the readers of your journal. Correspondence should be addressed to Prof. Gong at the following address, phone, and email address:

School of Management Science and Engineering

Shandong University of Finance and Economics,

Erhuandong Road,

Lixia District,

Jinan 250014, P.R. China.

Tel: (+86) 19861818113

E-mail: m1926017049@gmail.com

Thanks very much for your attention to my paper. We are looking forward to hearing from you soon.

Sincerely yours,

Pimei Gong

Detailed Response to Reviewers

Manuscript ID: [PONE-D-25-01170]

Paper Title: Corporate financial distress prediction with multiperiod annual report data: A fusion deep neural network model

Dear Editors and Reviewers,

Thanks for your letter and for the reviewers’ constructive comments concerning our manuscript entitled “Corporate financial distress prediction with multiperiod annual report data: A fusion deep neural network model” (Manuscript [PONE-D-25-01170]). These comments are all valuable and very helpful for improving our paper, as well as the important guiding significance to our research. We have modified our paper according to these comments carefully. We have used the "Track Changes" function in Microsoft Word to revise the manuscript without making any unmarked modifications. Please see purple colored parts for the revisions. We deeply hope that our revised paper could meet your requirements.

Section 1: a) and b)

a. The entire comments made by Reviewers

Reviewer #1:

The paper proposes a Fusion Deep Neural Network model based on Multiple annual Reports Text data and Financial data (MTF-FDNN). This model is capable of extracting long-text features from multi-period corporate annual reports as well as features from multi-period financial indicators. To further improve the manuscript, the following suggestions are given:

1、In the paper, some figures in the manuscript are a little blurry, please improve the clarity.

2、The formatting specifications throughout the manuscript need to be carefully checked and revised. For example, formulas need to be centered.

3、Since there are some papers in this topic, the contributions of the manuscript should be better summarized and listed.

4、While the introduction sets the context, a more explicit literature review section could better situate the study within the broader research landscape, such as Financial distress prediction with annual reports-based deep textual feature extraction: A hybrid approach, EVADE Targeted Adversarial False Data Injection Attacks for State Estimation in Smart Grid, and so on. These references could provide valuable insights into your research.

5、Add a section on the limitations of the work and future work in this paper.

Reviewer #2

The introduction section

1. The introduction could be strengthened by incorporating more recent literature to better contextualize the study. For example, the sentence: “Previous studies have shown that predictive models that combine textual and financial data can significantly increase the accuracy of financial distress predictions.” lacks citations for the referenced studies. Similarly, the authors state: “In the early stages of financial distress prediction research, scholars domestically and internationally relied primarily on financial indicators from a single fiscal year, applying machine learning or deep learning techniques for prediction. In recent years, text mining has become a hot topic in academia.” Again, no supporting references are provided. Including relevant citations would enhance the credibility of these claims.

2. A clearer transition from the problem statement to the proposed solution would improve the overall flow of the introduction.

3. The choice of Longformer, Bi-LSTM, and TextCNN requires stronger theoretical justification. Why is Longformer better than other transformers? Why were Bi-LSTM and TextCNN combined instead of alternative architectures (e.g., hierarchical attention networks)? Providing a rationale for these choices would strengthen the methodological foundation of the study.

4. Adding a brief description of the structure of the article at the end of the introduction would enhance its clarity and help guide the reader through the following sections.

The related works section

5. Some of the references cited in the manuscript are relatively old (e.g., Beaver, 1966). Additionally, the manuscript does not contain many sources, and most references are older than five years.

For example, you state “Hajek P. et al. [15] applied XGBoost and AdaBoost algorithms for classification analysis of samples that integrate textual metrics. García V. et al. [16] further improved the classification accuracy of models by combining ensemble strategies such as bagging and random subspace methods including more recent studies would provide a more current context.”, the mentioned authors, as well as others, have more recent studies in this subject area. Including newer literature would help provide a more up-to-date context.

6. The organization of related studies into subsections is appreciated, as it enhances the clarity of the article. However, the discussion of individual studies is too brief. The authors introduce the problems each study addresses but do not provide an in-depth discussion of the methodologies or results. A more detailed analysis of prior work, including quantitative performance metrics (e.g., accuracy rates of previous models), would improve the quality of this section. Furthermore, while the limitations of prior studies are mentioned, the research gaps they leave unaddressed are not explicitly stated. Identifying these gaps more clearly would help to better understand the described studies within The related works section.

The experiments and results section

7. The CSMAR dataset is used, but there is no justification for why this dataset is representative. More information about the dataset, such as the number of companies, the time span covered, and other relevant characteristics, should be provided. Additionally, there is no discussion of cross-validation or alternative training/test splits to ensure the model avoids overfitting. The authors should explain why they chose a 60:20:20 split and whether they experimented with other ratios (e.g., 70:20:10).

8. The comparison between traditional machine learning models and MTF-FDNN is clear, but the claims of superiority would be more robust with statistical significance testing (e.g., p-values). Adding such analyses would strengthen the validity of the results.

Conclusion and future works sections

9. The conclusion could be more reflective, discussing the broader impact and potential applications of the proposed model.

10. A deeper comparison between the proposed MTF-FDNN model and previous financial distress prediction models mentioned in the Related works section is missing. While the article presents results, it does not explicitly compare them to previous state-of-the-art models in financial distress prediction. The authors should clarify why MTF-FDNN is superior to BERT-based models, CNN-only models, or other hybrid architectures.

11. A discussion of computational efficiency and trade-offs between accuracy and interpretability would be beneficial. Addressing these aspects would provide a more balanced assessment of the model's practical implications.

12. There is no discussion including the limitations and implications of the research.

b. The Point-to-Point responses to the Reviewers.

Reply to Editors

We are grateful to you for providing us with the opportunity to strengthen the paper. The detailed explanations can be seen below.

Reviewer #1:

Comment 1: Please ensure that your manuscript meets PLOS ONE's style requirements, including those for file naming.

Response: We have revised the manuscript accordingly to ensure compliance with the journal’s guidelines, including formatting, file naming, and citation styles.

Comment 2: Please note that PLOS ONE has specific guidelines on code sharing for submissions in which author-generated code underpins the findings in the manuscript. In these cases, we expect all author-generated code to be made available without restrictions upon publication of the work.

Response: Additional explanation on code sharing was provided in the data availability section, and necessary code is available on GitHub (https://github.com/1926017049/MTF-FDNN-model).

Comment 3: Please state what role the funders took in the study. If the funders had no role, please state: "The funders had no role in study design, data collection and analysis, decision to publish, or preparation of the manuscript."

Response: The funders had no role in study design, data collection and analysis, decision to publish, or preparation of the manuscript and declaration has been added to the funding section.

Comment 4: PLOS requires an ORCID iD for the corresponding author in Editorial Manager on papers submitted after December 6th, 2016. Please ensure that you have an ORCID iD and that it is validated in Editorial Manager. To do this, go to ‘Update my Information’ (in the upper left-hand corner of the main menu), and click on the Fetch/Validate link next to the ORCID field. This will take you to the ORCID site and allow you to create a new iD or authenticate a pre-existing iD in Editorial Manager.

Response: We have ensured that the corresponding author has a registered and validated ORCID iD in Editorial Manager.

Thanks again for your kind reminder. We deeply hope that our revised paper could meet your requirements. Thank you very much again for your kind help.

Reply to Reviewer #1

We are grateful to you for identifying the weaknesses in our paper and providing us with the opportunity to strengthen the paper. The detailed explanations can be seen below.

Reviewer #1: The paper proposes a Fusion Deep Neural Network model based on Multiple annual Reports Text data and Financial data (MTF-FDNN). This model is capable of extracting long-text features from multi-period corporate annual reports as well as features from multi-period financial indicators.

Comment 1: In the paper, some figures in the manuscript are a little blurry, please improve the clarity.

Response: We have addressed the issue of blurry figures in the manuscript by replacing the affected images with high-resolution versions. Additionally, all figures were carefully checked and adjusted for sharpness to ensure consistency throughout the document.

Comment 2: The formatting specifications throughout the manuscript need to be carefully checked and revised. For example, formulas need to be centered.

Response: We have thoroughly revised the manuscript formatting according to the journal's guidelines, with specific attention to the issues you raised:

Formula Alignment: All formulas have been centered and consistently numbered.

Comment 3: Since there are some papers in this topic, the contributions of the manuscript should be better summarized and listed.

Response: In the introduction section, we have clearly summarized the main contributions of our manuscript. Specifically, our contributions include:

(1) Development of a feature fusion model to extract semantic features from long texts.

This study applies multiperiod financial report texts to the field of financial distress prediction and designs a novel neural network architecture tailored to the characteristics of multiperiod financial report texts. This approach enhances the prediction performance of financial distress models that incorporate long texts from multiperiod financial reports. First, the pretrained Longformer model is utilized to compute word vectors for the long texts in financial reports. Subsequently, the global features of the text were extracted via bidirectional long short-term memory (Bi-LSTM), and the local features were extracted via a Text Convolutional Neural Network (TextCNN). A feature-level fusion method is then used to construct multiperiod semantic features of long texts, offering a new perspective for extracting semantics from multiperiod financial report texts.

(2) A dynamic financial distress prediction model is constructed on the basis of multiperiod annual report data.

A dynamic financial distress prediction model is constructed on the basis of multimodal data from multiperiod annual report texts and financial indicators. First, multiple fully connected neural networks are used to process multiperiod financial indicator data. Next, a fusion model is designed to handle multiperiod annual report texts by extracting long texts from the Management Discussion and Analysis (MD&A) sections of multiple years. The feature fusion model is then used to compute the global and local semantic features of these long texts, and the results are integrated as multiperiod text semantic features. Finally, a neural network combines the extracted multiperiod textual semantics and multiperiod financial features for predictive computation.

To further clarify our contributions, we have refined the wording in the introduction to ensure that readers can better understand the novelty and significance of our work. We hope this revision addresses the reviewer's concern. Thank you for your insightful suggestion.

Comment 4: While the introduction sets the context, a more explicit literature review section could better situate the study within the broader research landscape, such as financial distress prediction with annual reports-based deep textual feature extr

---

## [Decision Letter · Decision Letter 1]

1 Aug 2025

PONE-D-25-01170R1Corporate financial distress prediction with multiperiod annual report data: A fusion deep neural network modelPLOS ONE

Dear Dr. Gong,

Thank you for submitting your manuscript to PLOS ONE. After careful consideration, we feel that it has merit but does not fully meet PLOS ONE’s publication criteria as it currently stands. Therefore, we invite you to submit a revised version of the manuscript that addresses the points raised during the review process.

Please carefully address the comments raised by the reviewers.

We look forward to receiving your revised manuscript.

Kind regards,

Arne Johannssen

Academic Editor

PLOS ONE

Journal Requirements:

Reviewers' comments:

Reviewer's Responses to Questions

**Comments to the Author**

1. If the authors have adequately addressed your comments raised in a previous round of review and you feel that this manuscript is now acceptable for publication, you may indicate that here to bypass the “Comments to the Author” section, enter your conflict of interest statement in the “Confidential to Editor” section, and submit your "Accept" recommendation.

Reviewer #1: (No Response)

Reviewer #2: All comments have been addressed

2. Is the manuscript technically sound, and do the data support the conclusions?

Reviewer #1: (No Response)

Reviewer #2: Yes

3. Has the statistical analysis been performed appropriately and rigorously? 

Reviewer #1: (No Response)

Reviewer #2: Yes

4. Have the authors made all data underlying the findings in their manuscript fully available?

Reviewer #1: (No Response)

Reviewer #2: Yes

5. Is the manuscript presented in an intelligible fashion and written in standard English?

Reviewer #1: (No Response)

Reviewer #2: Yes

6. Review Comments to the Author

Reviewer #1: Please revise carefully based on the comments raised. Although experimental results are provided, the paper would benefit from comparative analysis with current state-of-the-art approaches. More in depth comparison and analysis should be given in the manuscript.

Reviewer #2: The authors deal with an interesting topic. The topic is well work out at the required level in term of content and of formal aspect.

7. PLOS authors have the option to publish the peer review history of their article (what does this mean? ). If published, this will include your full peer review and any attached files.

**Do you want your identity to be public for this peer review?** For information about this choice, including consent withdrawal, please see our Privacy Policy .

Reviewer #1: No

Reviewer #2: No

---

## [Author Response · Author response to Decision Letter 2]

12 Aug 2025

Section 1: a) and b)

a. The entire comments made by Reviewers

Reviewer #1: Please revise carefully based on the comments raised. Although experimental results are provided, the paper would benefit from comparative analysis with current state-of-the-art approaches. More in depth comparison and analysis should be given in the manuscript.

Reviewer #2: The authors deal with an interesting topic. The topic is well work out at the required level in term of content and of formal aspect.

b. The Point-to-Point responses to the Reviewers.

Reply to Reviewer #1

We are grateful to you for providing us with the opportunity to strengthen the paper. The detailed explanations can be seen below.

Reviewer #1:

Please revise carefully based on the comments raised. Although experimental results are provided, the paper would benefit from comparative analysis with current state-of-the-art approaches. More in depth comparison and analysis should be given in the manuscript.

Response: Thank you for your valuable comments. We have added a new section, 4.4 "Parameter Sensitivity Analysis," in the manuscript, where we conduct a detailed analysis of several key parameters to further enrich the experimental results and discussion. Please refer to the main text for details. We hope these additions better demonstrate the advantages and performance of our method.

---

## [Decision Letter · Decision Letter 2]

9 Sep 2025

Corporate financial distress prediction with multiperiod annual report data: A fusion deep neural network model

PONE-D-25-01170R2

Dear Dr. Gong,

We’re pleased to inform you that your manuscript has been judged scientifically suitable for publication and will be formally accepted for publication once it meets all outstanding technical requirements.

Kind regards,

Arne Johannssen

Academic Editor

PLOS ONE

Additional Editor Comments (optional):

Reviewer #2:

Reviewers' comments:

Reviewer's Responses to Questions

**Comments to the Author**

1. If the authors have adequately addressed your comments raised in a previous round of review and you feel that this manuscript is now acceptable for publication, you may indicate that here to bypass the “Comments to the Author” section, enter your conflict of interest statement in the “Confidential to Editor” section, and submit your "Accept" recommendation.

Reviewer #2: All comments have been addressed

2. Is the manuscript technically sound, and do the data support the conclusions?

Reviewer #2: Yes

3. Has the statistical analysis been performed appropriately and rigorously? 

Reviewer #2: Yes

4. Have the authors made all data underlying the findings in their manuscript fully available?

Reviewer #2: Yes

5. Is the manuscript presented in an intelligible fashion and written in standard English?

Reviewer #2: Yes

6. Review Comments to the Author

Reviewer #2: I have reviewed the revised manuscript and the authors' responses to the comments from the second review. I am pleased to see that all the questions and concerns raised have been thoroughly addressed in their revisions.

7. PLOS authors have the option to publish the peer review history of their article (what does this mean? ). If published, this will include your full peer review and any attached files.

**Do you want your identity to be public for this peer review?** For information about this choice, including consent withdrawal, please see our Privacy Policy .

Reviewer #2: No

---

## [Editor Report · Acceptance letter]

PONE-D-25-01170R2

PLOS ONE

Dear Dr. Gong,

I'm pleased to inform you that your manuscript has been deemed suitable for publication in PLOS ONE. Congratulations! Your manuscript is now being handed over to our production team.

Kind regards,

on behalf of

Profesor Arne Johannssen

Academic Editor

PLOS ONE